# A Preliminary Study on the Interplay between the Serum Levels of Neurotransmitters and Thyroid Hormones for the Evaluation of the Behavioral Phenotype of Dogs

**DOI:** 10.3390/ani13030411

**Published:** 2023-01-26

**Authors:** Raffaella Cocco, Francesca Arfuso, Claudia Giannetto, Giuseppe Piccione, Alberto Cesarani, Giuseppe Pulina, Sara Sechi

**Affiliations:** 1Department of Veterinary Medicine, University of Sassari, 07100 Sassari, Italy; 2Department of Veterinary Sciences, University of Messina, 98168 Messina, Italy; 3Department of Agriculture, University of Sassari, 07100 Sassari, Italy; 4Department of Animal and Dairy Science, University of Georgia, Athens, GA 30602, USA

**Keywords:** behavior phenotype, neurotransmitters, dog breed, thyroid hormones

## Abstract

**Simple Summary:**

The domestic dog (*Canis familiaris*) represents an ideal model to study the effects of the selection process on motivation in companion animals. The dog’s temperament is related to breed and is controlled by neurotransmitter levels. Measuring these hormonal mediators can confirm or contradict the behavioral standards of the animal breed. The current study showed that in dog breeds classified with different behavioral standards, neurotransmitter values could reveal abnormal behaviors that cannot be assessed with simple ethograms.

**Abstract:**

A total of 112 dogs (49 males and 63 females) belonging to different breeds (i.e., Boxer, Cirneco dell’Etna, Fonni’s Dog, Labrador, Crossbreed, German Shepherd, Pit Bull, Shar-Pei, Yorkshire) were analyzed to compare the serum concentration of serotonin, dopamine, norepinephrine, prolactin, beta-endorphins, thyroxine (T4), triiodothyronine (T3), thyroid-stimulating hormone (TSH), and assess whether these parameters can be correlated with the behavioral phenotype of the investigated breeds. T4 was above or below the threshold in 61% and 14% of dogs, respectively; T3, in contrast, 41% of dogs showed values below the limit, while 26% above it. TSH was within the reference range in 58% of dogs; 94% of the dogs had prolactin in the reference range and only five animals showed values above the limit. For beta-endorphins, 49% of dogs had values above the limit, while 46% had values within the reference range. Serotonin and dopamine values below physiological limits were found in 62% and 70% of dogs, respectively. Finally, 61% of the dogs showed norepinephrine values within the reference range. The study confirmed that the assessment of the serum values of hormones and neurotransmitters in dogs could be useful to better understand the behavioral phenotype of the animal and could be useful for breeders and trainers for the selection of the most suitable subjects for specific tasks.

## 1. Introduction

Domestication and selection processes have shaped the morphological, productive and behavioral characteristics of agricultural animals (e.g., cattle, sheep, goats, pigs) and companion animals (e.g., dogs, horses) [1]. For both, selection acted more or less deeply on temperament, but for the former, the selective objectives were mainly on productive and reproductive traits, while for the latter, the selection was mainly concerned with personality and the ability to help and interact with human beings. The personality of animals is identified with individual consistency in behavioral reactivity to stimuli and situations [2]. The domestic dog (*Canis familiaris*) represents an ideal model for studying the effects of these processes in companion animals. There is a vigorous debate on the domestication of this species, and several studies have reported different locations, timing and a number of events [3,4,5]. The hypothesis that has received the greatest consensus is that this species originated from the domestication of wild gray wolves about 15 thousand years ago [6], although other studies hypothetically placed this event about 36 thousand years ago [7,8,9,10]. Saetre et al. [11] compared gene expression patterns in dogs, wolves and coyotes (a close relative), finding rapid changes in brain gene expression among these species and identifying genes with region-specific expression patterns in all species studied.

Moreover, it has been suggested that the strong selection of behavior that dogs underwent during the domestication process may have changed the patterns of mRNA expression in different genes of the hypothalamus [11], which is a brain region involved in different functions (e.g., hormone release, stimulation, control of feed intake, fear processing). In addition to its other functions, the hypothalamus plays an important role in behavioral responses [12]. Several important hormones are associated with hypothalamus-releasing factors: e.g., gonadotropin-releasing hormone, growth hormone-releasing hormone, thyrotropin-releasing hormone, corticotropin-releasing hormone, somatostatin, dopamine, oxytocin, vasopressin [13]. Though it could be hypothesized that the neurotransmitter levels in systemic circulation would mirror their physiological actions in the different brain regions, it is important to consider this issue a potential limitation when interpreting the results regarding neurotransmitter levels in the blood. Breeds such as Pit Bull, American Staffordshire, Bull Terrier, Yorkshire and Jack Russell show a high motivation for patrol/exploration, research, predatory, kinesthetic (moving), affiliative, territorial, protective, possessive, competitive, while they tend to have a low collaborative impulse, intra- and interspecific social, communicative, epimeletic (asking or giving care). Often these needs are not met, and for this reason, these dogs can show above-average reactivity and unpredictability, with thyroxine (T4), norepinephrine, and beta-endorphins above the reference range. The low values of these parameters could be associated with a lack of gratification and satisfaction, as demonstrated by the values of dopamine and serotonin below the threshold [14]. The Labrador breed (water rescue dogs) has a high motivation for patrol/exploration, search, predator, kinesthetic (moving), affiliated, cooperative, intra and interspecific social, syllegic (collecting objects and taking them to a hiding place), communicative, epimeletic (asking or giving care), while it tends to have a low territorial drive, protective, possessive, competitive (Standard n.122, 12 January 2011 International Kennel Federation, FCI, http://dev.fci.be/Nomenclature/Standards/122g08-en.pdf, accessed on 20 February 2021). These characteristics make them suitable for different social contexts and to be included in animal-assisted interventions and all research activities (civil and military protection). However, they are not suitable for the defense of territory and people. The consequences associated with dissatisfaction and, in some cases, poor selection can result in hyperactive dogs with high responsiveness, obsessive-compulsive disorder, and lack of attention [12,13,14].

It is known that changes in the number or function of neural crest cells can influence behavior because, for example, the adrenal and pituitary systems, which influence aggression and behavioral reactions, originated from these cells [15,16].

Nowadays, there are about 400 recognized dog breeds [17], selected in the past and now bred for different purposes. Based on their characteristics, the Italian National Dog Organization (ENCI) recognizes different groups of dogs: shepherd dogs, guard and defense dogs, different types of hunting dogs, primitive dogs, and companion dogs. The levels of hormones and neurotransmitters were modeled during this selection process; therefore, the hormone profiles are different between different breeds. Hormone and neurotransmitter levels may be associated with different behaviors of dog breeds [18], which show a genetic basis [19].

The aim of this study was to compare the serum concentrations of serotonin, dopamine, norepinephrine, prolactin, beta-endorphins, triiodothyronine (T3), thyroxine (T4) and thyroid-stimulating hormone (TSH) in different dog breeds, and to assess whether these parameters can be correlated with the motivations that drive dog breeds. The results gathered in the current study could give useful information for breeders and/or trainers; as a matter of fact, the deep knowledge of the relationship between the concentration of hormones and the presence of characteristic behaviors for a breed can help breeders and trainers to select the most suitable subjects for specific tasks. This would also lead to greater well-being for dogs, respecting their natural predispositions.

## 2. Materials and Methods

### 2.1. Animals, Sampling and Laboratory Analysis

A total of 112 dogs (49 males and 63 females) were recruited with the consent of their owners within a normal clinical routine. The study was carried out during Autumn 2020, and protocol of animal husbandry and experimentation was reviewed and approved in accordance with the standards recommended by the Guide for the Care and Use of Laboratory Animals and Directive 2010/63/EU for animal experiments. The animals belonged to 9 different breeds: Boxer (BOX, 3), Cirneco dell’Etna (CIR, 2), Fonni’s Dog (FON, 38), Labrador (LAB, 23), Crossbreed (CRO, 9), German Shepherd (GS, 18), Pit Bull (PIT, 14), Shar-Pei (SHA, 2) and Yorkshire (YOR, 3).

All dogs underwent a routine clinical and behavioral examination. From each dog, a blood sample was collected from the cephalic vein of the forearm into vacutainer tube containing EDTA (Terumo Corporation, Tokyo, Japan) and into vacutainer tube without anticoagulant agent (Terumo Corporation, Japan). Both tubes were centrifuged at 5232× g for 1.5 min at 37 °C at the same hour of the day (09:00–10:00 a.m.) in order to assess the serum concentrations of triiodothyronine (T3), thyroxine (T4); thyroid stimulating hormone (TSH), prolactin, serotonin, beta-endorphins and the plasma concentrations of dopamine and norepinephrine. Serum and plasma samples were stored at −20 °C prior to analysis, were tested by means of Crocodile instrument, mini Workstation, Totertek Berthold, with commercially available kits previously validated for canine species (Canine T3-ELISA kit, Bluegene (Shanghai, China), sensitivity of 1.0 ng/mL; T4 total ELISA kit, Demeditec Diagnostics GmbH (Kiel, Germany), sensitivity of 1.0 ng/mL; TSH Canine ELISA Demeditec Diagnostics GmbH, sensitivity of 0.01 ng/mL; Prolactin canine ELISA Demeditec Diagnostics GmbH, sensitivity of 0.4 ng/mL; Serotonin Research EIA Demeditec Diagnostics GmbH, Sensitivity of 0.005 ng/mL; Dopamine Plasma ELISA Demeditec Diagnostics GmbH, sensitivity of 7 pg/mL; Canine Beta-Endorfine Elisa Kit Bluegene, sensitivity of 1.0 ng/mL; Norepinephrine EIA Demeditec Diagnostics GmbH, sensitivity of 50 pg/mL) to evaluate the concentrations of the parameters mentioned above.

### 2.2. Description of the Breeds

All owners were asked to describe where the dog lived (apartment, house with garden, box), how many family members he had, how he spent his day, number of outings, what games he played and with whom if he played sports (utility and defense, agility, research). Moreover, the ethogram and behavioral form (Appendix A) were compiled for each subject to assess whether the behavioral characteristics were compatible with those described in the breed standard. Veterinarians, experts in analyzing animal behavior, confirmed the definition given by the owners based on observations made on dogs in different situations, approaching strangers, playing (ball, push and pull), and approaching other dogs. The main traits investigated were stranger-directed aggression, owner-directed aggression, dog-directed aggression/fear, trainability, chasing, stranger-directed fear, nonsocial fear, dog-directed fear, separation-related behavior, touch sensitivity, excitability, attachment, or attention-seeking. These behavioral traits were assessed using the C-Barq questionnaire. The hormonal and neurotransmitter differences in the investigated breeds were evaluated using the classification of the ENCI (Ente Nazionale della Cinofilia Italiana) group.

Labrador Retriever: Group 8 ENCI (Standard n.122, 12 January 2011 International Kennel Federation, FCI, http://dev.fci.be/Nomenclature/Standards/122g08-en.pdf, accessed on 20 February 2021). Employment: retriever dog. Behavior and character: good temperament, very agile. Excellent sense of smell, soft grip, and great passion for water. Devoted companion who knows how to adapt. Intelligent, passionate and helpful, with a great desire to be appreciated. Gentle by nature, he is never aggressive and inappropriately shy.

Boxer: Group 2 ENCI (FCI standard N° 144/9 July 2008, https://docslib.org/doc/3106344/fci-standard-n%C2%B0144-09-07-2008-gb-boxer-deutscher-boxer, accessed on 20 February 2021). Use: Utility and companion dog. Behavior and character: the Boxer must be firm of nerves, self-confident, calm and balanced. Its character is of the utmost importance and requires, in breeding, the utmost attention. Their attachment and loyalty to the owner and the whole house, his vigilance, and his indomitable courage as a defender have long been famous. It is harmless in the family but wary of strangers, gay and friendly in the game, but fearless in difficult situations. It is easy to train, given his docility, confidence and courage, natural acumen and extraordinary sense of smell. Moreover, being a clean dog with few needs, it is appreciated in the family both as a companion dog and as a defense or utility dog. Its character is loyal, without falsehood or dissimulation, and he maintains these qualities until old age.

Shar-Pei: Group 2 ENCI (FCI Standard N° 309/9 August 1999, http://www.fci.be/nomenclature/Standards/309g02-en.pdf, accessed on 20 February 2021). Use: hunting and guard dog. Behavior-character: calm, independent, faithful, and affectionate towards his family.

Canna Fonnese: Group 2 ENCI standard: breed not yet recognized FCI. USE: guard dog and guard dog for flocks and herds. Behavior and character: working dog, is never ruthless with the animals it takes care of, but it can be, if necessary, an excellent guardian against any predators (foxes or stray dogs). True to its rustic origins, it must be treated as a true companion you can always count on. It is necessary to know the dog species and appreciate its primitive rusticity, intelligence, dignity and strong guard instinct. If it does not work, they must perform adequate physical activity to preserve their innate balance.

Cirneco dell’Etna: Group 5 ENCI (FCI Standard n.199/30 October 2016, http://www.fci.be/nomenclature/Standards/199g05-en.pdf, accessed on 20 February 2021). Use: Hunting, especially of wild rabbits. Behavior and character: hunting dog, suitable for rough terrain and especially for hunting wild rabbits. A dog with a good character, at the same time sweet and affectionate.

German Shepherd Dog: Group 1 ENCI (FCI Standard N° 166/23 December 2010, http://www.fci.be/nomenclature/Standards/166g01-en.pdf, accessed on 20 February 2021). Usage: utility dog, sheepdog and service dog. Behavior and character: the German Shepherd must be balanced, constant, self-confident, accommodating, and (if not provoked) good, as well as attentive and docile. It must possess courage, combativeness and temperament to be suitable as a companion, guard, protection, service and shepherd dog.

Pit Bull: there is no single standard. The Pit Bull was born as a fighting dog; those who selected them wanted to be sure they would not be bitten when it was necessary to separate the two dogs during the fight. This characteristic, combined with the extraordinary affection that Pit Bulls have for their owners, makes them excellent companions. However, Pit Bulls are strong and impetuous dogs and very aggressive towards other dogs. These dogs need to move (walk, play, exercise), so they are not suitable for sedentary owners. They are not dogs with simple personalities. They are among the dogs for which a lot of experience is needed because they tend to be stubborn and impose themselves as dominant if the owner is too weak and uncertain [20].

Fonni’s Dog: there is no single standard. Preliminary research on Fonni’s dog has been carried out by the AACF (Fonni’s Dog Lovers Association), investigating the genetic variability of the breed and genetic overlaps with other recognized dog breeds sharing the same geographic region and behavioral traits. The history of Fonni’s dogs, together with their role as protectors of livestock and property and as hunters, is well documented. The ability of Fonni’s dog subjects to create a homogeneous breed-specific group has been suggested [21] though notable overlap does occur with the other Italian breeds selected for the same working abilities.

Yorkshire Terrier: Group 3 ENCI (FCI Standard N° 86/22 February 2012/EN, http://www.fci.be/nomenclature/Standards/086g03-en.pdf, accessed on 20 February 2021). Use: Pet terrier. Behavior and character: smart and intelligent companion dog. Lively and constantly in the mood. It was once used for hunting mice, hence its lively and cheerful character. He is not easily intimidated, neither by larger dogs nor by strangers, towards whom he shows a brazen determination. It is certainly very effective in surveillance and does not hesitate to signal with its barking something foreign that moves around it. At the same time, they can be particularly sweet and affectionate towards their owners, always active and sometimes even petulant. Like many terriers, it always claims to be the center of attention.

### 2.3. Statistical Analysis

All traits studied (i.e., stranger-directed aggression, owner-directed aggression, dog-directed aggression/fear, trainability, chasing, stranger-directed fear, nonsocial fear, dog-directed fear, separation-related behavior, touch sensitivity, excitability, attachment or attention-seeking) were compared with the values of serum and plasma parameters obtained from investigated dogs.

Pearson’s correlation analysis was performed between the levels of hormones and neurotransmitters studied to highlight their mutual relationship. The coefficients were declared significantly different from zero to *p* < 0.05.

The following linear model was applied to the variables studied:y = μ + A + S + B + e(1)
where: y was the variable investigated; μ was the overall average; A was the covariate of the dog’s age; S was the fixed effect of sex (2 levels, male and female); B was the fixed effect of the breed (9 levels, i.e., the breed mentioned above); and it was the term of error. The investigated variables were log-transformed before being statistically analyzed. Differences in means were reported to be significant when the *p* value of the ANOVA test was less than 0.05. whether the fixed effect considered was significant, the least squares averages were separated through the Tukey test, and significance was declared for *p* < 0.05. All statistical analyses were carried out in R software [22].

## 3. Results

Table 1 shows the average values of each considered parameter together with the reference values. The breeds with the highest frequency of the results beyond the reference intervals were Fonni’s dog, Labrador, German Shepherd and Pit Bull. The concentrations of TSH resulted within reference ranges in 58% of dogs. T4 was above and below the threshold in 61% and 14% of dogs, respectively; T3 values were below the limit in 41% of dogs and above in 26% of dogs. 94% of the dogs showed prolactin values within the reference range, whereas five dogs showed values above the ranges. 49% of the dogs had beta-endorphins values above the reference range, while the values of this parameter were within the reference range in 46% of investigated dogs. Serotonin and dopamine values were found below the reference ranges in 62% and 70% of dogs, respectively. Finally, 68% of the dogs showed norepinephrine values within the reference range.

The sample size of four dog breeds herein investigated (i.e., Boxer, Cirneco dell’Etna, Shar-Pei, Yorkshire) was less than five animals; thus, the data obtained from these breeds were excluded from the statistical analysis. The overall correlations between the parameters studied are shown in Table 2. Interesting mild and positive correlations were highlighted between dopamine and serotonin (0.41, *p* < 0.001), norepinephrine and serotonin (0.57, *p* < 0.001), and norepinephrine and dopamine (0.65, *p* < 0.001). The age and sex of the animals were never significant for the parameters investigated (Table 3). The breed significantly affected the values of beta-endorphins, dopamine, T3 and T4 (*p* < 0.001). Specifically, dogs belonging to Crossbreed showed higher beta-endorphins and T3 values (*p* < 0.001) than dogs belonging to Labrador, German Shepherd, Pit Bull and Fonni’s dog breeds. Labrador showed higher beta-endorphins, dopamine and T4 values than Crossbreed, German Shepherd, Pit Bull and Fonni’s dog (*p* < 0.001).

## 4. Discussion

The dog’s character is related to breed and is controlled by neurotransmitter and hormonal levels. Therefore, measuring these chemical mediators can confirm or contradict the behavioral standards of the animal breed. In the present study, the serum concentrations of thyroid hormones and neurotransmitters were assessed in dogs belonging to different breeds in order to assess their usefulness in revealing abnormal behaviors that cannot be assessed with simple ethograms. The results herein found that the highest frequency of data beyond the reference intervals for investigated parameters was found in the breeds Fonni’s dog, Labrador, German Shepherd and Pit Bull. Moreover, 32% of dogs showed norepinephrine values outside the reference range.

Animals with TSH values within reference ranges show that these animals have not been affected by hypo- or hyper-thyroidism, while changes in T3 and T4 values are probably related to adrenal activity due to environmental stimuli. A total of 13 Labrador dogs (56.5%) showed T4 levels above the physiological level. The increase in T4 leads to an increase in the body’s sensitivity to catecholamines. T3 increases the speed and amplitude of peripheral nerve reflexes, alertness, responsiveness to various stimuli, memory and learning ability [23]. Breed-related differences were found in T3 and beta-endorphins, with higher values recorded in dogs belonging to Crossbreed compared to dogs belonging to Labrador, German Shepherd, Pit Bull and Fonni’s dog breeds, whereas dogs belonging to Labrador breed showed higher dopamine and T4 values than Crossbreed, German Shepherd, Pit Bull and Fonni’s dog.

A previous study carried out on male dogs with apparently normal fertility showed that potential breed differences in the secretion of hormones related to the pituitary–gonadal and –thyroid axis as confirmed by changes in circulating concentrations of prolactin and thyroid hormones including TSH and T4 [24]. Thyroid hormones appear to affect the concentrations of serotonin in the blood and different brain regions and modulate serotonin turnover in the brain [25,26]. Involvement of the serotonergic system in dogs with aggression has been established in several reports [25,27,28]. An interbreed variation in serum serotonin concentration was found in healthy dogs aged between 1 and 7 years [29]. Canine reference values are lacking, but higher serum serotonin concentrations have been found in healthy Cavalier King Charles Spaniels (CKCS) dogs compared with healthy dogs of other breeds [30]. Another study carried out on healthy dogs showed that Scottish Terriers, Maltese, and Belgian Shepherds had higher serum serotonin concentrations than other terrier breeds and Labrador Retrievers [31]. In the current study, only seven dogs showed serotonin values above the reference ranges: increased values may be related to intense exercise and state within social groups.

Serotonin is defined as the hormone of good mood. Low levels of serotonin seem to be associated with intra- and interspecific aggression, anxiety and depression. Low dopamine values (found in 78 dogs) could be associated with a reduced ability of the subject to cope with stress, aggression during stress and easy distraction, as well as loss of interest in normally pleasurable activities [14,32]. Only 5% of the dogs had beta-endorphins below the normal threshold, demonstrating a condition of anxiety and chronic stress. Beta-endorphins, synthesized in the pituitary gland, adrenal glands and certain tracts of the digestive system, are released during a strong emotion or in particularly stressful situations to help the individual better tolerate pain by influencing mood [33].

Regarding the ENCI motivation grid, German Shepherds usually show a high propensity for kinesthetic, territorial, protective, possessive, competitive, and predatory motivation, but also research, collaboration and exploration. Thus, this breed is flexible and able to work side by side with humans in numerous activities. However, the German Shepherd breed can show problematic behaviors, expressive alterations and compensation mechanisms when their interaction with humans does not meet certain parameters, that is, if the motivations that require channeling (through disciplinary activities) do not receive the right attention. In this case, we can observe the most varied drifts in their behavior. Physical and/or psychological frustration can turn German Shepherds into introverted, exaggeratedly territorial, aggressive, or even compulsive animals. In the current study, three of 18 German Shepherd dogs showed aggressive behavior, and the same dogs also showed dopamine and serotonin values below the threshold and values related to stress (i.e., norepinephrine and beta-endorphins) above the threshold suggesting a relationship between the levels of neurotransmitters and the aggressive behavior as previously shown [14,34,35,36].

Regarding the behavior characteristics, one of these three dogs showing aggressive behavior was a black/tan male German Shepherd, aged 7 years old, living in an apartment with three adults. The dog went out of the house for a walk of about 20 min three times a day to fulfill his physiological needs. The dog did not play and/or did not engage in any physical activity. The dog was always in a state of alert in the house and barked at the slightest noise. On a walk, he pulled on the leash, and the owners couldn’t let him go because he didn’t return to the call and tried to bite all the people and dogs he met. The second dog showing aggressive behavior was a male gray German Shepherd, male, aged three years old, living in a box. Dog went out of the box to perform his physiological needs and to carry out utility and defense training lasting about half an hour twice a week. The training only involved coercive techniques. The dog had contact with the owner only twice a day when the food was delivered. In the box, he turned on himself chasing his tail. The dog was very quick in responding to the owner’s requests, towards whom he performed continuous pacifying and submissive behaviors. The dog was aggressive toward other dogs and towards strangers, especially children. The third dog showing aggressive behavior was a male black/tan German Shepherd, aged 6 years old, living in a house with a garden. The dog was always alone in the garden, and the owners did not have a play relationship with the dog. The dog spent the whole day running along the fence, chasing every car or person that passed. The dog did not answer the owner’s call and started spinning around, chasing his tail. The dog was aggressive towards strangers and owners, and other dogs. The dog was responsible for an attack on one of the owners, causing his jaw to break.

A total of three dogs (black/fairy, “beauty line”) from the same breeder had all values within the reference range, associated with a balanced behavioral profile and did not show any problematic behavior. Another dog (gray, “line of work”) with all the considered parameters above the threshold values carried out an intense physical activity (usefulness and defense, research). The investigated parameters suggested dogs were satisfied and gratified but difficult to manage due to the expression of a hyper type from the behavioral point of view. Boxer and Fonni’s dog turn out to be particularly different in their needs, clearly due to the different type of breeding, which often combines with the context of life. Though the data obtained from dogs belonging to these breeds were not included in the statistical analysis due to the low sample size of the group, it has been observed that of the three Boxers considered, the only one with values that showed full satisfaction and gratification was the dog that carries out research activities, the other two, while not showing behavioral problems, had values below the threshold for dopamine and serotonin, while the values of norepinephrine and beta-endorphins were above the threshold, confirming the description of exuberant and reactive dogs by owners and veterinarians expert in animal behavior assessment. Fonni’s dog comes from Sardinia, the second-largest island in the Mediterranean. Recent genetic studies have shown that this isolation is reflected in its genome, as demonstrated for humans [37]. Of the 38 Fonni’s dogs, 28 lived in sheep farms with the role of guardians and were always tied to the chain. Ten dogs lived in the house with a garden. A total of 21 of 38 (55.3%) Fonni’s dogs had T4 above range, while eight dogs (21.1%) showed levels below range. Nine of 38 dogs (23.7%) showed above-range endorphin levels, while 20 dogs (52.6%) showed low serotonin levels and 25 dogs (65.8%) showed low dopamine levels.

Among Labrador dogs investigated in the current study, one dog was brought to visit because he is hyperactive and difficult to manage in the family context. The results of his tests showed values above the threshold for T3, T4, beta-endorphins, and norepinephrine, demonstrating intense activation of the hypothalamic–pituitary–adrenal axis. However, dopamine in the reference range has shown an individual feeling gratified because the positive relationship with the owner satisfies his affiliative, social, and epimeletic needs. At the same time, serotonin below the reference threshold has highlighted the lack of satisfaction with activities that lead to a “good mood”, which can be represented in a physical activity aimed at satisfying the motivations of exploration, research, and predatory. Two Labradors purchased by the owner for beauty shows had values below the threshold for serotonin, dopamine, TSH and T3, while norepinephrine and beta-endorphins were normal. Due to a lack of gratification in race motivations, one of these manifested aggressive responses towards strangers, as demonstrated by low serotonin, which may be associated with aggressive antisocial behavior [38,39]. Dogs with serotonin and dopamine values with reference ranges are those that carry out research activities, which seem to be, in all the breeds evaluated, the activity able to satisfy dogs in their exploratory/exploratory motivations and to reduce many behavioral drifts related to frustration, boredom and under stimulation [14].

A limitation of this study was the small number of dogs of some breeds herein considered. Specifically, the results obtained from dogs belonging to four breeds herein investigated (i.e., Boxer, Cirneco dell’Etna, Shar-Pei and Yorkshire) should be interpreted carefully because of the small number of dogs herein tested; thus, further studies on the breeds herein investigated by enrolling a larger number of dogs are demanded in order to clarify the possible relationship between the serum levels of neurotransmitters and thyroid hormones and the behavioral phenotype of dogs. Another limitation to be acknowledged is related to ELISA kits not specific to dogs used in the present study. The kits were validated in our laboratory. However, we suggest that the validation of the same kits should be performed by each other laboratory which would use them.

## 5. Conclusions

The results gathered in the current study suggest the usefulness of the investigation of serum concentrations of thyroid hormones and neurotransmitters to better understand the behavioral phenotype of a dog’s breed. Knowing the relationship between the concentration of hormones and the presence of characteristic behaviors for a breed can help breeders and trainers to select the most suitable subjects for specific tasks. This would also lead to greater well-being for dogs, respecting their natural predispositions.

## Figures and Tables

**Table 1 animals-13-00411-t001:** Observed mean values for serum and plasma parameters, together with the reference ranges, obtained from investigated dogs.

Parameters	Mean ± SD	Reference Range ^1^	Threshold		Breed ^2^								
					BOX	CIR	FON	LAB	CRO	GS	PIT	SHA	YOR
Prolactin (ng/mL)	06.00 ± 06.07	0.73–16.65	Under	2 (2%)						1		1	
			Above	5 (4%)			3			1	1		
Beta-endorphins (ng/mL)	11.3 ± 21.10	0.4–0.6	Under	6 (5%)			2	3					1
			Above	55 (49%)	2	1	14	8	9	11	8	2	
T3 (ng/mL)	3.00 ± 15.99	0.5–5.9	Under	46 (41%)			24	11	1	2	5		3
			Above	29 (26%)	2		3	7	8	5	2	2	
T4 (ng/dL)	9.54 ± 16.3	5–12.5	Under	68 (61%)		1	29	6	6	13	8	2	3
			Above	16 (14%)	2			13			1		
TSH (ng/mL)	01.70 ± 2.66	0.7–2.8	Under	32 (29%)	1	1	11	7	3	7	2		
			Above	15 (13%)			6		2	4	3		
Serotonin (ng/mL)	11.5 ± 11.60	12–32.5	Under	69 (62%)	2		19	13	4	16	13	1	1
			Above	7 (6%)			2	4					1
Dopamine (pg/mL)	10.66 ± 10.30	0.3–20	Under	78 (70%)	2		36	10	6	12	10		2
			Above	8 (7%)			1	6				1	
Norepinephrine (ng/mL)	96.44 ± 10.40	0.07–4.21	Under	6 (5%)			3	1	1		1		
			Above	35 (31%)	1	2	9	8	2	4	5	2	2

^1^ Internal laboratory reference ranges. ^2^ BOX = Boxer; CIR = Cirneco dell’Etna; FON = Fonni’s dog; LAB = Labrador; CRO = Crossbreed; GS = German Shepherd; PIT = Pit Bull; SHA = Shar-Pei; YOR = Yorkshire.

**Table 2 animals-13-00411-t002:** Correlations (above the diagonal) and their coefficient (below the diagonal, *p*-value) between the variables considered.

Parameters	Prolactin	Beta-Endorphins	T3	T4	TSH	Serotonin	Dopamine	Norepinephrine
Prolactin		0.06	−0.06	0.08	0.11	−0.14	−0.11	0.04
Beta-endorphins	n.s.		0.47	−0.01	0.02	0.17	0.16	0.30
T3	n.s.	*p* < 0.001		0.02	0.10	0.15	0.27	0.37
T4	n.s.	n.s.	n.s.		−0.14	−0.02	−0.05	−0.07
TSH	n.s.	n.s.	n.s.	n.s.		−0.10	0.00	0.07
Serotonin	n.s.	n.s.	n.s.	n.s.	n.s.		0.41	0.57
Dopamine	n.s.	n.s.	*p* < 0.001	n.s.	n.s.	*p* < 0.001		0.65
Norepinephrine	n.s.	*p* < 0.001	*p* < 0.001	n.s.	n.s.	*p* < 0.001	*p* < 0.001	

n.s.—not significant.

**Table 3 animals-13-00411-t003:** Results of the linear model obtained for each parameter obtained from investigated dog breeds.

Parameters	ANOVA			Sex		Breed ^1^				
	Age	Sex	Breed	Female	Male	FON	LAB	CRO	GS	PIT
Prolactin	n.s.	n.s.	n.s.	6.36	5.6	6.96	3.80	6.22	5.43	8.10
Beta-endorphins	n.s.	n.s.	*p* < 0.001	8.39	15.03	8.62 ^a^	10.54 ^b^	33.40 ^c^	8.60 ^a^	8.52 ^a^
T3	n.s.	n.s.	*p* < 0.001	2.89	3.49	1.43 ^a^	4.04 ^a^	7.33 ^b^	3.60 ^a^	2.21 ^a^
T4	n.s.	n.s.	*p* < 0.001	10.83	7.87	3.63 ^b^	23.5 ^c^	4.43 ^b^	4.13 ^b^	6.64 ^b^
TSH	n.s.	n.s.	n.s.	1.77	1.62	1.38	1.00	1.40	3.46	2.20
Serotonin	n.s.	n.s.	n.s.	11.95	10.95	12.96	16.23	11.45	6.70	4.74
Dopamine	n.s.	n.s.	*p* < 0.001	3.73	3.57	0.86 ^a^	11.55 ^b^	1.15 ^a^	1.74 ^a^	0.32 ^a^
Norepinephrine	n.s.	n.s.	n.s.	4.93	4.99	3.77	9.57	2.61	3.13	3.06

^1^ FON—Fonni’s dog; LAB—Labrador; CRO—Crossbreed; GS—German Shepherd; PIT—Pit Bull; n.s.—not significant. Values in the same row of different superscripts are significantly different (*p* < 0.001).

## Data Availability

The data presented in this study are available on request from the corresponding author.

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
