# Peer review of "A Preliminary Study on the Interplay between the Serum Levels of Neurotransmitters and Thyroid Hormones for the Evaluation of the Behavioral Phenotype of Dogs"

_animals, 2023, doi:10.3390/ani13030411_

Round 1
Reviewer 1 Report
-The title clearly describes the article.
-The abstract has described the content of the manuscript. I suggest to delete the last sentence “This would also lead to greater well-being of dogs respecting their natural predispositions.”
-The introduction has explicated extensively on the problems being investigated and the manuscript could serve as an addition to the existing literature. However, there are some little English language and grammar mistakes to be corrected in the manuscript. Please, check and correct them.
Please change the sentence “These authors also concluded that the strong selection of behavior that dogs underwent during the domestication process may have changed the patterns of mRNA expression in different genes of the hypothalamus, which is a brain region involved in different functions (e.g., hormone release, stimulation, control of feed intake, fear processing). In addition to its other functions, the hypothalamus plays an important role in behavioral responses [12] and controls the pituitary, which in turn regulates various glands and endocrine organs.” as “Moreover, it has been suggested that the strong selection of behavior that dogs underwent during the domestication process may have changed the patterns of mRNA expression in different genes of the hypothalamus [11], which is a brain region involved in different functions (e.g., hormone release, stimulation, control of feed intake, fear processing). In addition to its other functions, the hypothalamus plays an important role in behavioral responses [12].”
-The methodology section has illustrated extensively on the method used. However I suggest some changes. In particular, I suggest to avoid the use “He, it or they” in the subsection “2.2 Motivation of breed” to refer to the breed of dog.
-The result section has been demonstrated by comprehensively.
In table 3 Authors wrote “Noradrenaline”, please change with “Norepinephrine”
-The discussion section is well expounded and described the manuscript clearly.
-The Conclusion is clear, concise, and brings a suggestion on how the knowledge acquired in this study could be used to clarify other matters.
-The reference list is reasonably comprehensive and relevant.
-Tables are generally good, and well represent the findings of the study.
Author Response
Dear Editor and Reviewers,
Thank you very much for reviewing our manuscript No. animals-2171540 entitle “Interplay between the serum levels of neurotransmitters and thyroid hormones for the evaluation of the behavior phenotype of dogs”
We have addressed all Reviewers’ concerns with as much detail as possible.
We have provided our detailed responses below and have edited our manuscript accordingly.
We outlined every change made throughout the manuscript according to the comments and suggestions of Reviewers.
We hope that our revised manuscript will be acceptable for publication in Animals.
Reviewers' comments and Authors’ responses:
Reviewer 1
The title clearly describes the article.
-We thank Reviewer for the positive comments.
-The abstract has described the content of the manuscript. I suggest to delete the last sentence “This would also lead to greater well-being of dogs respecting their natural predispositions.”
-We thank Reviewer for the positive comments. We deleted the sentence according to Reviewer suggestion.
-The introduction has explicated extensively on the problems being investigated and the manuscript could serve as an addition to the existing literature. However, there are some little English language and grammar mistakes to be corrected in the manuscript. Please, check and correct them.
-We thank Reviewer for the positive comments. We improved English language throughout the text.
Please change the sentence “These authors also concluded that the strong selection of behavior that dogs underwent during the domestication process may have changed the patterns of mRNA expression in different genes of the hypothalamus, which is a brain region involved in different functions (e.g., hormone release, stimulation, control of feed intake, fear processing). In addition to its other functions, the hypothalamus plays an important role in behavioral responses [12] and controls the pituitary, which in turn regulates various glands and endocrine organs.” as “Moreover, it has been suggested that the strong selection of behavior that dogs underwent during the domestication process may have changed the patterns of mRNA expression in different genes of the hypothalamus [11], which is a brain region involved in different functions (e.g., hormone release, stimulation, control of feed intake, fear processing). In addition to its other functions, the hypothalamus plays an important role in behavioral responses [12].”
-Done.
-The methodology section has illustrated extensively on the method used. However I suggest some changes. In particular, I suggest to avoid the use “He, it or they” in the subsection “2.2 Motivation of breed” to refer to the breed of dog.
-Done.
-The result section has been demonstrated by comprehensively.
-We thank Reviewer for the positive comments.
In table 3 Authors wrote “Noradrenaline”, please change with “Norepinephrine”
-We corrected it.
-The discussion section is well expounded and described the manuscript clearly.
-We thank Reviewer for the positive comments.
-The Conclusion is clear, concise, and brings a suggestion on how the knowledge acquired in this study could be used to clarify other matters.
-We thank Reviewer for the positive comments.
-The reference list is reasonably comprehensive and relevant.
-We thank Reviewer for the positive comments.
-Tables are generally good, and well represent the findings of the study.
-We thank Reviewer for the positive comments.
Reviewer 2 Report
Dear Authors,
The idea of the study is very interesting, with the implications for animal welfare and, in a further context, for the evolutionary aspects of interactions between genetic and environmental factors. The elaboration of the design is satisfactory and not complicated to follow. Nevertheless, several issues appeared. The first one related to the correlation between neurotransmitter levels in the blood and their actions in the brain tissue. Further, the laboratory methodology description and inclusion criteria for breeds have questionable items. The Results section needs updates and improvements. A thorough reorganization of the Discussion would reduce the impression of being unacceptably speculative and better acknowledge the scientific novelty. Please find below the detailed comments and suggestions.
Title and Introduction
- The title goes beyond the methodology, which did not involve the experimental evaluation of the behavior traits in studied dogs. Therefore, please consider rephrasing the title to indicate the potential of your results in further studies on canine behavior.
- An issue appeared related to the hypothesis. Namely, how confident was it assuming that the neurotransmitter levels in systemic circulation would mirror their physiological actions in the different regions of the brain? Please consider additional explanations supported by references.
- The text in lines 49−74 might be shorter, emphasizing the genetic traits in brain tissue.
- What were the criteria for the selection of the measured parameters? Did previous studies address the breed-associated differences in their levels? The answer to this question would better highlight the novelty of your results. Finally, what was the reason for omitting adrenalin since you measured the other two catecholamines?
Material and Methods
- It seems necessary to mention the study design and period. The suitability of the procedure regarding ethical approval does not seem complete. The research purpose analysis of the surplus of the samples collected for routine procedures is ethically acceptable. Nevertheless, the corresponding Ethical Committee for Animal Welfare should have approved the study due to the clinical features.
- Lines 96−8: Please consider the description indicating that the study included dogs of 8 breeds and nine crossbreed dogs. The suitability of pitbull inclusion seems questionable. The text in Lines 147−57 explained the behavior features of that breed, but there was no link with the ENCI standards.
- Laboratory methodology might seem like a source of significant issues. The serum is not suitable for noradrenaline and dopamine measurement; instead, plasma is the proper sample. Accordingly, the Manufacturer of the ELISA kits mentioned plasma and urine for dopamine and EDTA plasma with sodium metabisulfite for noradrenaline. Therefore the methodology should include validation procedures confirming the reliability of serum results. Further, the Manufacturer of the Prolactin kit in the Instructions for use does not seem to specify the applicability for a canine specimen. The website of the corresponding Manufacturer does not provide data about the canine T4 and TSH ELISA kits. The recommendation of validation also refers to these three parameters. Additionally, it might be necessary to check-up whether the quoted sensitivity data are valid. The coefficient of variation should be part of the assay specifications. The details about centrifugation (speed, duration) and storage (temperature, duration) conditions are missing.
- Please include the reference or web address where the ENCI standards are available. If presented in Table, the data in Lines 113−65 might be more convenient to follow. Also, please make sure to use uniform nomenclature for the breeds. For example, information is lacking for Fonni's Dog, mentioned in Lines 96−8.
- Line 168−9: Please cite the source of the reference ranges. Instead of threshold, reference interval might be a more suitable term.
- Line 173−81: Please explain the statistical methodology for testing the normality of distribution. In addition, it remained elusive whether the calculation parameters of the breed-specific models, presented in Table 3, needed additional correlation analyzes. Related to that, please explain the statistical correction made due to the repeated testing.
Results
- An additional table with the demographic and clinical data would improve the presentation of the results.
- When describing the content of Table 1, please consider additional efforts to avoid repetition. Further, it seems interesting to indicate the breeds with the highest frequency of the results beyond the reference intervals. Finally, please check the frequency of dogs with noradrenaline results outside the reference interval.
- In Table 1, please replace the physiological range with the reference range.
- Lines 198−200: The adjective mild would be more suitable for describing the correlations. The coefficient of determination (squared correlation coefficient), which represents the intensity of a relationship, would be between 17% and 42%.
- The legend for Table 2 should explain the abbreviation n.s. and the reason for marking some values as bolded.
- The content of Table 3 needs updates. The coefficient values should have a uniform number of decimal places. Furthermore, the interchangeable usage of the terms race and breed does not seem suitable, while the term badly introduced confusion. The legend should explain what the letters a−c stand for.
Discussion
- Overall, the comments are general and supported by an unacceptably small number of references. In line with that, please consider reducing the amount of breed-associated data and citing the corresponding references.
- The first paragraph should also summarize the main findings of the study.
- The results do not present data about the commented behavioral and environmental features, thus marking their correlates with the lab results as speculative. What might be an improvement is the comparison with the previous studies assessing breed-associated differences in thyroid hormones and neurotransmitters and the eventual relationship with behavioral traits.
- There are comments about the indications for the veterinary checkup, like in Lines 288−90, which reinforce the need for the Ethical Committee approval. In a similar context, please make additional efforts that all patient data remain confidential, which is currently not the case in Lines 288, 297, and 300.
- The final part of the discussion should acknowledge the study's limitations.
Conclusion
- The last concluding remark is beyond the methodological scope and without reliable support from the results. Therefore, please consider omitting it from the revised Manuscript.
Technical Suggestions
- Additional language editing would further improve the overall quality of the Manuscript.
- Please consider additional check-ups on spelling the names in Latin.
- Information on the vacuum tubes Manufacturer should be part of Materials and Methods.
Author Response
Dear Editor and Reviewers,
Thank you very much for reviewing our manuscript No. animals-2171540 entitle “Interplay between the serum levels of neurotransmitters and thyroid hormones for the evaluation of the behavior phenotype of dogs”
We have addressed all Reviewers’ concerns with as much detail as possible.
We have provided our detailed responses below and have edited our manuscript accordingly.
We outlined every change made throughout the manuscript according to the comments and suggestions of Reviewers.
We hope that our revised manuscript will be acceptable for publication in Animals.
Reviewers' comments and Authors’ responses:
Reviewer 2
Dear Authors,
The idea of the study is very interesting, with the implications for animal welfare and, in a further context, for the evolutionary aspects of interactions between genetic and environmental factors. The elaboration of the design is satisfactory and not complicated to follow. Nevertheless, several issues appeared. The first one related to the correlation between neurotransmitter levels in the blood and their actions in the brain tissue. Further, the laboratory methodology description and inclusion criteria for breeds have questionable items. The Results section needs updates and improvements. A thorough reorganization of the Discussion would reduce the impression of being unacceptably speculative and better acknowledge the scientific novelty. Please find below the detailed comments and suggestions.
-We sincerely thank Reviewer for his/her valuable revision which lead us to improve our manuscript. We considered all comments and suggestions and we work to satisfied them.
Title and Introduction
- The title goes beyond the methodology, which did not involve the experimental evaluation of the behavior traits in studied dogs. Therefore, please consider rephrasing the title to indicate the potential of your results in further studies on canine behavior.
-We thank for the suggestion. We added information on ethograms used for the behavior traits in studied dogs. Moreover, we changed the title as “A preliminary study on the interplay between the serum levels of neurotransmitters and thyroid hormones for the evaluation of the behavioral phenotype of dogs”
- An issue appeared related to the hypothesis. Namely, how confident was it assuming that the neurotransmitter levels in systemic circulation would mirror their physiological actions in the different regions of the brain? Please consider additional explanations supported by references.
-We thank for the comment and suggestion. We added more information on the topic in the introduction section. We wrote “…it has been suggested that the strong selection of behavior that dogs underwent during the domestication process may have changed the patterns of mRNA expression in different genes of the hypothalamus [11], which is a brain region involved in different functions (e.g., hormone release, stimulation, control of feed intake, fear processing). In addition to its other functions, the hypothalamus plays an important role in behavioral responses [12]. Several important hormones are associated with hypothalamus releasing factors: e.g., gonadotropin-releasing hormone, growth hormone releasing hormone, thyrotropin-releasing hormone, corticotropin-releasing hormone, somatostatin, dopamine, oxytocin, vasopressin [13]. Breeds such as Pitbull, American Staffordshire, Bull Terrier, Yorkshire and Jack Russell show a high motivation for patrol / exploration, research, predatory, kinesthetic (moving), affiliative, territorial, protective, possessive, competitive, while they tend to have a low collaborative impulse, intra- and interspecific social, communicative, epimeletic (asking or giving care). Often these needs are not met, and, for this reason, these dogs can show above-average reactivity and unpredictability, with T4, norepinephrine, beta-endorphins above the normal range. The low values of these parameters could be associated with lack of gratification and satisfaction as demonstrated by the values of dopamine and serotonin below the threshold [14]. The Labrador breed (water rescue dogs), has a high motivation for patrol / exploration, search, predator, kinesthetic (moving), affiliated, cooperative, intra and interspecific social, syllegic (collecting objects and taking them to a hiding place), communicative, epimeletic (asking or giving care), while it tends to have a low territorial drive, protective, possessive, competitive. These characteristics make them suitable for different social contexts and to be included in Animal Assisted Interventions and in all research activities (civil and military protection). However, they are not suitable for the defense of territory and people. The consequences associated with dissatisfaction and, in some cases, poor selection, can result in hyperactive dogs with high responsiveness, obsessive compulsive disorder, and lack of attention.”
- The text in lines 49−74 might be shorter, emphasizing the genetic traits in brain tissue.
-We thank Reviewer for the comment and suggestion. We shortened the paragraph and emphasized the genetic traits in brain tissue and we added, according to the suggestions of the other Reviewers, information related to behavior characteristics and relationship with neurotransmitters and hormones in dog’s breeds.
- What were the criteria for the selection of the measured parameters? Did previous studies address the breed-associated differences in their levels? The answer to this question would better highlight the novelty of your results.
-We thank Reviewer for the comment. We added more information on this topic with related references.
Material and Methods
- It seems necessary to mention the study design and period. The suitability of the procedure regarding ethical approval does not seem complete. The research purpose analysis of the surplus of the samples collected for routine procedures is ethically acceptable. Nevertheless, the corresponding Ethical Committee for Animal Welfare should have approved the study due to the clinical features.
-We thank Reviewer for the comments. We added missing information.
- Lines 96−8: Please consider the description indicating that the study included dogs of 8 breeds and nine crossbreed dogs. The suitability of pitbull inclusion seems questionable. The text in Lines 147−57 explained the behavior features of that breed, but there was no link with the ENCI standards.
-We understand Reviewer concern. However, as we specified in the text, there is no single standard for Pit Bull breed. As a matter of facts, the Ente Nazionale della Cinofilia Italiana (ENCI) does not recognize the American Pit Bull Terrier as a breed and therefore does not register its pedigrees, however in Italy there are associations that protect the Pitbull through their own registers, regulations and standards of race.
- Laboratory methodology might seem like a source of significant issues. The serum is not suitable for noradrenaline and dopamine measurement; instead, plasma is the proper sample. Accordingly, the Manufacturer of the ELISA kits mentioned plasma and urine for dopamine and EDTA plasma with sodium metabisulfite for noradrenaline. Therefore the methodology should include validation procedures confirming the reliability of serum results. Further, the Manufacturer of the Prolactin kit in the Instructions for use does not seem to specify the applicability for a canine specimen. The website of the corresponding Manufacturer does not provide data about the canine T4 and TSH ELISA kits. The recommendation of validation also refers to these three parameters. Additionally, it might be necessary to check-up whether the quoted sensitivity data are valid. The coefficient of variation should be part of the assay specifications. The details about centrifugation (speed, duration) and storage (temperature, duration) conditions are missing.
-We thank Reviewer for the comments and suggestions which allowed us to correct mistake related to blood collection and to clarify some aspects on serum and plasma analysis in the methods section. Moreover, We indicated details about centrifugation characteristics. The T4 and TSH kits used in our study are registered for mammalian species.
- Please include the reference or web address where the ENCI standards are available. If presented in Table, the data in Lines 113−65 might be more convenient to follow. Also, please make sure to use uniform nomenclature for the breeds. For example, information is lacking for Fonni's Dog, mentioned in Lines 96−8.
-We thank Reviewer for the comments and suggestions. We added the web address where ENCI standards are available as well as we added information and related reference for Fonni’s dog.
- Line 168−9: Please cite the source of the reference ranges. Instead of threshold, reference interval might be a more suitable term.
-We thank reviewer for the suggestion. We changed threshold with “reference intervals”. We specified in table that we used internal laboratory reference ranges as in literature no reference ranges for investigated parameters were available for healthy dogs when we performed the study.
- Line 173−81: Please explain the statistical methodology for testing the normality of distribution. In addition, it remained elusive whether the calculation parameters of the breed-specific models, presented in Table 3, needed additional correlation analyzes. Related to that, please explain the statistical correction made due to the repeated testing.
- We thank Reviewer for the question. We used the Pearson correlations (we now specified if in the text). According to your suggestion, we checked the distributions of our variables and then we decided to run again the model by log-transforming the variables to meet the assumptions (we checked them now). Thank you again for your suggestion.
Results
- When describing the content of Table 1, please consider additional efforts to avoid repetition. Further, it seems interesting to indicate the breeds with the highest frequency of the results beyond the reference intervals. Finally, please check the frequency of dogs with noradrenaline results outside the reference interval.
-We thank Reviewer for the valuable suggestions. We added the required information and we checked the frequency of dogs with noradrenaline results outside the reference interval and, thus, we corrected in the results section the sentence “…61% of the dogs showed norepinephrine values within the normal range.” with “…68% of the dogs showed norepinephrine values within the normal range.”
- In Table 1, please replace the physiological range with the reference range.
-Done.
- Lines 198−200: The adjective mild would be more suitable for describing the correlations. The coefficient of determination (squared correlation coefficient), which represents the intensity of a relationship, would be between 17% and 42%.
-We sincerely thank Reviewer for the suggestion. We changed “moderate” with “mild”.
- The legend for Table 2 should explain the abbreviation n.s. and the reason for marking some values as bolded.
-Done.
- The content of Table 3 needs updates. The coefficient values should have a uniform number of decimal places. Furthermore, the interchangeable usage of the terms race and breed does not seem suitable, while the term badly introduced confusion. The legend should explain what the letters a−c stand for.
-We thank Reviewer for the suggestions. We modified Table accordingly and we specified what the letters mean. We changed the term race with breed and badly with male.
Discussion
- Overall, the comments are general and supported by an unacceptably small number of references. In line with that, please consider reducing the amount of breed-associated data and citing the corresponding references.
-We thank Reviewer for the comments and we understand his/her concern. There are scant data on this topic. However, we checked for bibliographic information and we added some references on the investigated topic.
- The first paragraph should also summarize the main findings of the study.
-We thank Reviewer for the suggestion. We added this paragraph.
- The results do not present data about the commented behavioral and environmental features, thus marking their correlates with the lab results as speculative. What might be an improvement is the comparison with the previous studies assessing breed-associated differences in thyroid hormones and neurotransmitters and the eventual relationship with behavioral traits.
-We sincerely thank Reviewer for the comment and suggestions which help us to improve the discussion of the results of our study. According with Reviewer’s comment we added more insights on differences in thyroid hormones and neurotransmitters and the eventual relationship with behavioral trait in dogs.
- There are comments about the indications for the veterinary checkup, like in Lines 288−90, which reinforce the need for the Ethical Committee approval. In a similar context, please make additional efforts that all patient data remain confidential, which is currently not the case in Lines 288, 297, and 300.
-We sincerely thank Reviewer for the comment and suggestions. We specified in the methods section that protocol of animal husbandry and experimentation were reviewed and approved in accordance with the standards recommended by the Guide for the Care and Use of Laboratory Animals and Directive 2010/63/EU for animal experiments, and we specified that all patient data remain confidential. Therefore, we deleted all information on patient ID from Lines 288, 297 and 300.
- The final part of the discussion should acknowledge the study's limitations.
-We agree with Reviewer’s suggestion. We added a last paragraph acknowledging the study's limitations. We wrote “A limitation of this study was the small number of dogs of some breeds herein considered. Specifically, the results obtained from dogs belonging to 4 breeds herein investigated (i.e. Boxer, Cirneco dell’Etna, Shar-Pei and Yorkshire) should be interpreted carefully because of the small number of dogs herein tested; thus, further studies on the breeds herein investigated by enrolling a larger number of dogs are demanded in order to clarify the possible relationship between the serum levels of neurotransmitters and thyroid hormones and the behavioral phenotype of dogs.”
Conclusion
- The last concluding remark is beyond the methodological scope and without reliable support from the results. Therefore, please consider omitting it from the revised Manuscript.
-We thank Reviewer for the suggestion. We deleted the last sentence.
Technical Suggestions
- Additional language editing would further improve the overall quality of the Manuscript.
-We thank for the valuable suggestion. We checked and improved the overall quality of the Manuscript
- Please consider additional check-ups on spelling the names in Latin.
-We thank for the valuable suggestion. We checked and improved the names in Latin.
- Information on the vacuum tubes Manufacturer should be part of Materials and Methods.
-We added information on vacuum tubes.
Reviewer 3 Report
Review for Animals manuscript:
“Interplay between the serum levels of neurotransmitters and thyroid hormones for the evaluation of the behavior phenotype of dogs”
I have several comments to make about this manuscript. Most importantly, the authors need to clarify certain aspects, particularly if and which behavioral tests were performed and the statistics performed, re-structure the introduction to provide essential background information (incl. references) on which they base their hypotheses and predictions (which are currently missing), and caution about making general statements of breeds when only a very small number of animals was tested (e.g., N=3 for Boxers).
Here are my comments:
Major comments:
Line 96-98: I am worried about seeing that for almost half of the studied breeds, the sample size was actually less than 5 animals (e.g., Boxer, Cirneco dell Etna, Sharpei, Yorkshire). I think this precludes you from being able to infer anything meaningful for these breeds in general and it may make sense to exclude those from your analyses or at least be very careful about your conclusions for these breeds.
Line 99: I am glad to see that the dogs all underwent a behavioral examination. This was not clear from reading either the summary or abstract, so I would recommend that you clearly state this above already. Please also describe the type and nature of the behavioral examination. Was this a standardized test or did you design a test for this study?
Line 113-165: I am not sure whether the heading “motivation of the breed” is the right word – wouldn’t “classification” or “descriptions of the breeds” be more applicable? The whole paragraph should be restructured to avoid listing the breeds one after the other with sometimes just single word descriptions. Consider putting the information into a table or use separate paragraphs for each breed. It would also be good to have a consistent style describing each breed. Currently, you are using just single words for some breeds (e.g. Shar Pei) but thorough descriptions for others (e.g. Boxer, Pit Bull). In general, I don’t think the descriptions are very clear in terms of what the behavioral phenotype really looks like in real – life. If I understand correctly, these are the “ideals” as postulated in the classification of the dog breed associations. Can you provide the detailed ethograms / forms you used to assess the dogs mentioned in line 115?
I also would recommend not using the word “master” but rather “owner” or “caretaker”.
Line 168: You haven’t yet described which “traits” you actually tested. This is related to the comment above – please provide a description of the behavioral assessments you performed before continuing with statistics. – by having continued to read the manuscript, I realize that by traits you refer only to the hormone/neurotransmitter levels. So in fact, did you run any behavioral tests at all?
Line 167-181: Please provide more details on your statistical approach. It is not clear for instance, which correlation analysis (Spearman? Pearson? …) and modelling approach you took (full-null model comparison performed to avoid multiple testing? See for reference, Forstmeier, W., & Schielzeth, H. (2011). Cryptic multiple hypotheses testing in linear models: overestimated effect sizes and the winner's curse. Behavioral ecology and sociobiology, 65(1), 47-55.), whether the assumptions for your linear model were met (normally distributed and homogenous residuals for example), whether you log transformed the response (this is commonly done with hormonal concentrations to meet assumptions), or variables (z-transformation of co-variates for example), tests of collinearity, stability and confidence intervals are also not described. Further, I don’t understand why you include the overall average as a predictor which is also not reported in Table 3.
Line 220-222: You state that the neurotransmitters/hormones you measure are related to behavior but I don’t see anywhere in the paper (and this should be in the introduction) so far, what exactly your predictions are in this regard and whether there is anything already published on it (as far as I am aware, thyroid hormones have been associated with aggression in dogs but this is not mentioned thus far). This is severely lacking and leaves the reader confused and wondering what you are exactly testing in this study.
Line 224: What “abnormal” behaviors? These have to be defined somewhere.
Line 236-242: Please provide relevant references for these statements.
Line 252-254: “In the current study of 3 of 18 GS showed aggressive behavior, as demonstrated by dopamine and serotonin values below the threshold and values related to stress (i.e. norepinephrine and beta-endorphins) above the threshold” – do I understand correctly, that you did not actually test the behavior of these dogs? Or if you did, please provide the details and results of this assessment. Otherwise, you cannot claim that they show aggression based on physiological results. You can merely state that this combination of neurotransmitter levels may be related to aggressive behavior (if this has been described before in other studies? Provide a reference).
Related to this, line 266-267: “…confirming the description of exuberant and reactive dogs”: who provided this description of the dogs?
Line 270-287: I think this information about the connection of breed-specific behaviors and neurotransmitter levels should be moved to the introduction. This paragraph is not about discussing your results but rather necessary background information that should be available to the reader earlier on.
Minor comments:
1) Title: behavioral phenotype instead of behavior phenotype
2) Abstract: is crossing a dog breed or do you refer to crossbreeds (mixed breed dogs)? Related to that, if you refer to mixed breed dogs, does this not by definition exclude them from the current study where you state that you want to investigate breed-related differences?
3) Abstract: T4 was above or below instead of above and below
4) TSH was within normal range
5) Line 50: positions should be locations
6) Line 53: rejected should be placed?
7) Line 71: T should be t (testing)
8) Line 95: what does annualized mean in this context?
9) Line 99: a blood sample
10) Line 100: please clarify what you mean by whey vacutainer?
11) Line 208: meaning should be replaced by correlation coefficient
12) Table 3: Parameters – race should be breed, badly should presumably be male? Also, it is not clear what the different letters mean. Please provide footnotes. In addition – what are the numbers you report in this table? The model estimates for each predictor, or the means? Please clarify.
13) Line 228: 13 of out how many Labradors? It would be helpful to indicate either the % or the total number of animals in brackets. This applies to all further statements in the discussion.
14) Discussion: it would help the reader if you spelled out the breed names rather than using the abbreviations.
15) Line 247: what do you mean by “file”?
16) Line 267-269: Can you provide info on their hormonal profile as you did with the other breeds?
17) Line 274: reasons should be needs or drives?
18) Line 276: do you have a reference for this statement?
Author Response
Dear Editor and Reviewers,
Thank you very much for reviewing our manuscript No. animals-2171540 entitle “Interplay between the serum levels of neurotransmitters and thyroid hormones for the evaluation of the behavior phenotype of dogs”
We have addressed all Reviewers’ concerns with as much detail as possible.
We have provided our detailed responses below and have edited our manuscript accordingly.
We outlined every change made throughout the manuscript according to the comments and suggestions of Reviewers.
We hope that our revised manuscript will be acceptable for publication in Animals.
Reviewers' comments and Authors’ responses:
Reviewer 3
“Interplay between the serum levels of neurotransmitters and thyroid hormones for the evaluation of the behavior phenotype of dogs”
I have several comments to make about this manuscript. Most importantly, the authors need to clarify certain aspects, particularly if and which behavioral tests were performed and the statistics performed, re-structure the introduction to provide essential background information (incl. references) on which they base their hypotheses and predictions (which are currently missing), and caution about making general statements of breeds when only a very small number of animals was tested (e.g., N=3 for Boxers).
Here are my comments:
Major comments:
Line 96-98: I am worried about seeing that for almost half of the studied breeds, the sample size was actually less than 5 animals (e.g., Boxer, Cirneco dell Etna, Sharpei, Yorkshire). I think this precludes you from being able to infer anything meaningful for these breeds in general and it may make sense to exclude those from your analyses or at least be very careful about your conclusions for these breeds.
-We thank Reviewer for his/her valuable comments and suggestions. Accordingly, we excluded the breeds characterized by the small sample size from the statistical analysis of data and we clarify this aspect in results as well as in discussion section.
Line 99: I am glad to see that the dogs all underwent a behavioral examination. This was not clear from reading either the summary or abstract, so I would recommend that you clearly state this above already. Please also describe the type and nature of the behavioral examination. Was this a standardized test or did you design a test for this study?
-We thank Reviewer for the comments. We underwent a behavioral examination by a means of the Canine Behavioral Assessment and Research Questionnaire (C BARQ), a validated and reliable instrument for assessing dogs’ typical and recent responses to a variety of common stimuli and situations. We added this questionnaire as supplementary file S1. All owners were asked to describe where the dog lived (apartment, house with garden, box), how many family members he had, and how he spent his day, number of outings, what games he played and with whom, if he played sports (utility and defense, agility, research). We specified these information in the text.
Line 113-165: I am not sure whether the heading “motivation of the breed” is the right word – wouldn’t “classification” or “descriptions of the breeds” be more applicable? The whole paragraph should be restructured to avoid listing the breeds one after the other with sometimes just single word descriptions. Consider putting the information into a table or use separate paragraphs for each breed. It would also be good to have a consistent style describing each breed. Currently, you are using just single words for some breeds (e.g. Shar Pei) but thorough descriptions for others (e.g. Boxer, Pit Bull). In general, I don’t think the descriptions are very clear in terms of what the behavioral phenotype really looks like in real – life. If I understand correctly, these are the “ideals” as postulated in the classification of the dog breed associations. Can you provide the detailed ethograms / forms you used to assess the dogs mentioned in line 115?
-We thank Reviewer for the comments and suggestions. We modified the title of subsection as “Description of breed” as suggested. We provided the ethogram / form used as supplementary files S1. Moreover, we better organize the paragreaph in order to make it more clear for the reader.
I also would recommend not using the word “master” but rather “owner” or “caretaker”.
-Done
Line 168: You haven’t yet described which “traits” you actually tested. This is related to the comment above – please provide a description of the behavioral assessments you performed before continuing with statistics. – by having continued to read the manuscript, I realize that by traits you refer only to the hormone/neurotransmitter levels. So in fact, did you run any behavioral tests at all?
-We thank Reviewer for the comment and suggestion. We specified the tested traits in the text (stranger-directed aggression, owner-directed aggression, dog-directed aggression/fear, trainability, chasing, stranger-directed fear, nonsocial fear, dog-directed fear, separation-related behavior, touch sensitivity, excitability, attachment or attention-seeking).
Line 167-181: Please provide more details on your statistical approach. It is not clear for instance, which correlation analysis (Spearman? Pearson? …) and modelling approach you took (full-null model comparison performed to avoid multiple testing? See for reference, Forstmeier, W., & Schielzeth, H. (2011). Cryptic multiple hypotheses testing in linear models: overestimated effect sizes and the winner's curse. Behavioral ecology and sociobiology, 65(1), 47-55.), whether the assumptions for your linear model were met (normally distributed and homogenous residuals for example), whether you log transformed the response (this is commonly done with hormonal concentrations to meet assumptions), or variables (z-transformation of co-variates for example), tests of collinearity, stability and confidence intervals are also not described. Further, I don’t understand why you include the overall average as a predictor which is also not reported in Table 3.
-We thank Reviewer for the comments and suggestions. We used the Pearson correlations (we now specified if in the text). According to your suggestion, we checked the distributions of our variables and then we decided to run again the model by log-transforming the variables to meet the assumptions (we checked them now).
Line 220-222: You state that the neurotransmitters/hormones you measure are related to behavior but I don’t see anywhere in the paper (and this should be in the introduction) so far, what exactly your predictions are in this regard and whether there is anything already published on it (as far as I am aware, thyroid hormones have been associated with aggression in dogs but this is not mentioned thus far). This is severely lacking and leaves the reader confused and wondering what you are exactly testing in this study.
-We added more insights on this topic in introduction section as well as in discussion section with related references.
Line 224: What “abnormal” behaviors? These have to be defined somewhere.
-We thank Reviewer for the comment and suggestion. We defined it throughout the text.
Line 236-242: Please provide relevant references for these statements.
-Done.
Line 252-254: “In the current study of 3 of 18 GS showed aggressive behavior, as demonstrated by dopamine and serotonin values below the threshold and values related to stress (i.e. norepinephrine and beta-endorphins) above the threshold” – do I understand correctly, that you did not actually test the behavior of these dogs? Or if you did, please provide the details and results of this assessment. Otherwise, you cannot claim that they show aggression based on physiological results. You can merely state that this combination of neurotransmitter levels may be related to aggressive behavior (if this has been described before in other studies? Provide a reference).
-We thank Reviewer for the comment and suggestions. We added the missing information on the 3 GS showing aggressive behavior. We added references on relationship between the levels of neurotransmitters and the aggressive behavior in dog.
Related to this, line 266-267: “…confirming the description of exuberant and reactive dogs”: who provided this description of the dogs?
-We thank Reviewer for the comment and suggestion. We specified in the text “Veterinarians expert in analyzing animal behavior have confirmed the definition given by the owners, based on observations made on dogs in different situations, approaching strangers, playing (ball, push and pull), approaching other dogs.
Line 270-287: I think this information about the connection of breed-specific behaviors and neurotransmitter levels should be moved to the introduction. This paragraph is not about discussing your results but rather necessary background information that should be available to the reader earlier on.
-We thank Reviewer for the suggestion. We moved these information in the introduction section.
Minor comments:
1) Title: behavioral phenotype instead of behavior phenotype
-Done
2) Abstract: is crossing a dog breed or do you refer to crossbreeds (mixed breed dogs)? Related to that, if you refer to mixed breed dogs, does this not by definition exclude them from the current study where you state that you want to investigate breed-related differences?
-We thank Reviewer for the comment and suggestion. We refereed to crossbreeds. We corrected it throughout the text. We did not exclude this breed from statistical analysis since it would be interesting to understand how the parameters studied vary in dogs belonging to crossbreeds as they are very numerous among the dog population in Italy. However, if the reviewer deems it appropriate we will remove this group from the statistical analysis.
3) Abstract: T4 was above or below instead of above and below
-Done.
4) TSH was within normal range
-We changed it as suggested.
5) Line 50: positions should be locations
-We changed it as suggested.
6) Line 53: rejected should be placed?
-We changed it with placed.
7) Line 71: T should be t (testing)
-Done.
8) Line 95: what does annualized mean in this context?
-It was a mistake. We deleted the word.
9) Line 99: a blood sample
-Done.
10) Line 100: please clarify what you mean by whey vacutainer?
-We corrected as “tube without anticoagulant agent”
11) Line 208: meaning should be replaced by correlation coefficient
-Done.
12) Table 3: Parameters – race should be breed, badly should presumably be male? Also, it is not clear what the different letters mean. Please provide footnotes. In addition – what are the numbers you report in this table? The model estimates for each predictor, or the means? Please clarify.
-We thank Reviewer for the suggestions. The numbers reported in the table are the coefficient of correlation. We improved Table according to suggestions and comments of the Reviewer and we specified what the letters mean.
13) Line 228: 13 of out how many Labradors? It would be helpful to indicate either the % or the total number of animals in brackets.
-We thank Reviewer for the suggestion. We indicate the percentage as suggested (13 out of 23 Labrador, 56.5%).
14) Discussion: it would help the reader if you spelled out the breed names rather than using the abbreviations.
-Done
15) Line 247: what do you mean by “file”?
-We corrected the mistake. Thank you for pointing tis out to us.
16) Line 267-269: Can you provide info on their hormonal profile as you did with the other breeds?
We thank Reviewer for the suggestions. We added the information “Of the 38 Fonni’s dogs, 28 lived in sheep farms with the role of guardians and were always tied to the chain. Ten dogs lived in the house with a garden. A total of 21 of 38 (55.3%) Fonni’s dogs had T4 above range, while 8 dogs (21.1%) showed levels below range. Nine of 38 dogs (23.7%) showed above-range endorphin levels while 20 dogs (52.6%) showed low serotonin levels and 25 dogs (65.8%) showed low dopamine levels.”
17) Line 274: reasons should be needs or drives?
-Done.
18) Line 276: do you have a reference for this statement?
-We added reference for the statement as suggested.
Round 2
Reviewer 2 Report
Dear Authors,
The thorough improvements in the overall quality of the Manuscript are evident. Nevertheless, there are issues for which further improvements seem due. Please consider the following comments and suggestions.
Introduction
- The text in Lines 64−80 did not completely resolve the hypothesis-related concern−confidence in the assumption that the neurotransmitter levels in systemic circulation would mirror their physiological actions in the different brain regions. Please, consider this issue a potential limitation or precaution when interpreting the results.
- Lines 72−80: The reference(s) should support the descriptions.
Materials and Methods
- Please provide the names of the Ethical Committee endorsed to review the documentation. It is usual practice to include the number of the permission.
- The specificities of the Pit-Bull breed seem objective. However, please cite the relevant source(s) supporting the included characteristics.
- The serum and plasma storage conditions are lacking. Also, please quote in centrifugation speed in g instead of rpm.
- The species specificity issues related to the ELISA kits persisted. The prolactin kit remained without any comments. If you validated the T4 and TSH ELISA kits (prolactin, TSH, and T4), please include the results. Otherwise, please cite the literature source providing the validation data that the ELISA kit used in your study is suitable for the canine specimen. In neither of these is an option, please acknowledge these issues as the limitations necessitating caution in interpretation.
- Please consider additional efforts to uniformly use the term reference range throughout the Manuscript.
Results
- A significant amount of repetition remained in Lines 226−34.
Technical suggestions
- Please consider the Additional proofreading. For example, it might be more suitable than he in lines 196−201.
- Please separate the limitations as a separate paragraph.
Author Response
Comments and Suggestions for Authors
Dear Authors,
The thorough improvements in the overall quality of the Manuscript are evident. Nevertheless, there are issues for which further improvements seem due. Please consider the following comments and suggestions.
-We sincerely thank Reviewer for his/her revisions that help us to improve our manuscript. We considered all your new comments and suggestions.
Introduction
- The text in Lines 64−80 did not completely resolve the hypothesis-related concern−confidence in the assumption that the neurotransmitter levels in systemic circulation would mirror their physiological actions in the different brain regions. Please, consider this issue a potential limitation or precaution when interpreting the results.
-We thank Reviewer for the comment. We added this concern in the introduction section as suggested.
- Lines 72−80: The reference(s) should support the descriptions.
-Done.
Materials and Methods
- Please provide the names of the Ethical Committee endorsed to review the documentation. It is usual practice to include the number of the permission.
-We added the section “Ethical review and approval were exempted for this study, as the analyzed samples and animal behavior are derived from routine examinations that the dogs underwent at the veterinary medical clinic of the University of Sassari. The dog owners were informed and gave their consent to use their dogs' data for this work.”
- The specificities of the Pit-Bull breed seem objective. However, please cite the relevant source(s) supporting the included characteristics.
-We specified that there is no single standard for Pit Bull. However, we added a reference as Reviewer suggested.
- The serum and plasma storage conditions are lacking. Also, please quote in centrifugation speed in g instead of rpm.
-Done
- The species specificity issues related to the ELISA kits persisted. The prolactin kit remained without any comments. If you validated the T4 and TSH ELISA kits (prolactin, TSH, and T4), please include the results. Otherwise, please cite the literature source providing the validation data that the ELISA kit used in your study is suitable for the canine specimen. In neither of these is an option, please acknowledge these issues as the limitations necessitating caution in interpretation.
-We thank Reviewer for the comment and suggestions. We added the acknowledgment in the limitation study paragraph.
- Please consider additional efforts to uniformly use the term reference range throughout the Manuscript.
-Done.
Results
- A significant amount of repetition remained in Lines 226−34.
-We improved the section.
Technical suggestions
- Please consider the Additional proofreading. For example, it might be more suitable than he in lines 196−201.
-We thank Reviewer for the suggestion. We perform suggested correction throughout the text.
- Please separate the limitations as a separate paragraph.
-Done.
Reviewer 3 Report
The authors have substantially improved the manuscript and addressed my comments. I have some minor and one major comment still:
Major:
Statistical analyses: I am not sure how to understand the first sentence. What do you mean by physiological ranges in relation to behavioural traits? This needs clarification. In particular, I would like an explanation of how exactly the first sentence relates to the model formula below: What is y and what do you mean by the overall average (of what?)?
It is also confusing that in Table 1 the word “traits” is used to describe neurohormone concentrations. Please ensure consistent terminology throughout the manuscript.
Minor:
Line 133-135: I appreciate the further details the authors provided. But I still need clarifying: the veterinary expert performed observations on all these behaviours during the visit, is this correct? If this is the case, how did you test the behaviours, e.g., how was trainability tested? How was non-social fear assessed? And so on. Otherwise please indicate that these behavioural traits were just assessed using the C-Barq questionnaire.
Line 141-201: This paragraph could still be changed for improved readability by using subheadings, bullet points or a table.
Table 2 heading: Correlation coefficients are indicated above the diagonal, but below you are showing the p-value and not a coefficient, correct?
Author Response
Comments and Suggestions for Authors
The authors have substantially improved the manuscript and addressed my comments. I have some minor and one major comment still:
-We sincerely thank Reviewer for his/her revisions that help us to improve our manuscript. We considered all your new comments and suggestions.
Major:
Statistical analyses: I am not sure how to understand the first sentence. What do you mean by physiological ranges in relation to behavioural traits? This needs clarification. In particular, I would like an explanation of how exactly the first sentence relates to the model formula below: What is y and what do you mean by the overall average (of what?)?
-We thank Reviewer for the questions. We rewrote the sentence as “All traits studied (i.e. stranger-directed aggression, owner-directed aggression, dog-directed aggression/fear, trainability, chasing, stranger-directed fear, nonsocial fear, dog-directed fear, separation-related behavior, touch sensitivity, excitability, attachment or attention-seeking) were compared with the values of serum and plasma parameters obtained from investigated dogs. ”
It is also confusing that in Table 1 the word “traits” is used to describe neurohormone concentrations. Please ensure consistent terminology throughout the manuscript.
-We thank for the suggestions. We corrected the mistake.
Minor:
Line 133-135: I appreciate the further details the authors provided. But I still need clarifying: the veterinary expert performed observations on all these behaviours during the visit, is this correct? If this is the case, how did you test the behaviours, e.g., how was trainability tested? How was non-social fear assessed? And so on. Otherwise please indicate that these behavioural traits were just assessed using the C-Barq questionnaire.
-We thank Reviewer for the suggestion. We specified that these behavioural traits were just assessed using the C-Barq questionnaire.
Line 141-201: This paragraph could still be changed for improved readability by using subheadings, bullet points or a table.
-We thank Reviewer for the valuable suggestion. We improved it by starting the description of each breed with another paragraph.
Table 2 heading: Correlation coefficients are indicated above the diagonal, but below you are showing the p-value and not a coefficient, correct?
-We thank Reviewer for the suggestion. Yes it is correct. We clarify it in Table.